# The CBS test: Development, evaluation & cross-validation of a community-based injury severity scoring system in Cameroon

Haley Tupper[1]*, Rasheedat Oke[1], Catherine Juillard[1], Fanny Dissak-DeLon[2], Alain Chichom-Mefire[3], Mbiarikai Agbor Mbianyor[1], Georges Alain Etoundi-Mballa[4], Thompson Kinge[5], Louis Richard Njock[6], Daniel N. Nkusu[7], Jean-Gustave Tsiagadigui[8], Melissa Carvalho[1], Mark Yost[1], S. Ariane Christie[1]

1 Department of Surgery, Program for the Advancement of Surgical Equity (PASE), University of California Los Angeles, Los Angeles, California, United States of America, 2 Faculty of Health Sciences, University of Bamenda, Bamenda, Cameroon, 3 Faculty of Health Sciences, University of Buea, Buea, Cameroon, 4 Department of Disease Epidemic and Pandemic Control, Ministry of Health, Yaoundé, Cameroon, 5 Hospital Administration, The Limbe Regional Hospital, Lime, Cameroon, 6 Hospital Administration, The Laquintinie Hospital of Douala, Douala, Cameroon, 7 Hospital Administration, The Catholic Hospital of Pouma, Pouma, Cameroon, 8 Hospital Administration, The Edea Regional Hospital, Edea, Cameroon

* htupper@mednet.ucla.edu

**Data Availability Statement:** The de-identified data is available in S1 Data.

## Abstract

Injury-related deaths overwhelmingly occur in low and middle-income countries (LMICs). Community-based injury surveillance is essential to accurately capture trauma epidemiology in LMICs, where one-third of injured individuals never present to formal care. However, community-based studies are constrained by the lack of a validated surrogate injury severity metric. The primary objective of this bipartite study was to cross-validate a novel community-based injury severity (CBS) scoring system with previously-validated injury severity metrics using multi-center trauma registry data. A set of targeted questions to ascertain injury severity in non-medical settings–the CBS test—was iteratively developed with Cameroonian physicians and laypeople. The CBS test was first evaluated in the community-setting in a large household-based injury surveillance survey in southwest Cameroon. The CBS test was subsequently incorporated into the Cameroon Trauma Registry, a prospective multi-site national hospital-based trauma registry, and cross-validated in the hospital setting using objective injury metrics in patients presenting to four trauma hospitals. Among 8065 surveyed household members with 503 injury events, individuals with CBS indicators (CBS +) were more likely to report ongoing disability after injury compared to CBS- individuals (OR 1.9, p = 0.004), suggesting the CBS test is a promising injury severity proxy. In 9575 injured patients presenting for formal evaluation, the CBS test strongly predicted death in patients after controlling for age, sex, socioeconomic status, and injury type (OR 30.26, p<0.0001). Compared to established injury severity scoring systems, the CBS test comparably predicts mortality (AUC: 0.8029), but is more feasible to calculate in both the community and clinical contexts. The CBS test is a simple, valid surrogate metric of injury severity that can be deployed widely in community-based surveys to improve estimates of injury severity in under-resourced settings.

**Funding:** Dr. S. Ariane Christie received salary support from the Association for Academic Surgery Global Surgery Resident Scholarship (2017) ($30,000 to SAC). The Cameroon Trauma Registry was supported by Dr. Catherine Juillard's departmental research funding from University of California San Francisco from 2017-2019 (CJ). The funders had no role in study design, data collection and analysis, decision to publish, or preparation of the manuscript.

**Competing interests:** The authors have declared that no competing interests exist.

## Introduction

Globally, 8% of mortality is injury-related [1] and 90% of these deaths occur in low and middle-income countries (LMICs) [2]. For every injury-related death, an additional 20–50 people will sustain a non-fatal life-altering injury, resulting in disability and financial hardship [3,4]. It is imperative to accurately identify the burden of injury to appropriately direct limited resources and to design efficient care systems. Most LMIC injury epidemiology is derived from hospital-based registries, resulting in gross underestimates of the burden of trauma. One-third of injured individuals never receive formal medical care [5,6] and among those who do receive formal care, many only present to outpatient clinics [5]. Community-based injury surveillance is essential to accurately capture trauma epidemiology in LMICs.

Community-based studies in under-resourced settings are constrained by the lack of a validated method to assess injury severity. Injury severity proxies employed in high-income countries, including hospitalization, post-injury disability, and death, are significantly complicated by treatment access in lower-resource environments; many non-medical factors influence utilization of formal medical care (hospitalization) and post-injury disability is confounded by receipt of appropriate medical treatment [6]. Validated injury severity scores all rely on objective physiologic or anatomic variables unavailable in the community setting. Personal perception of one's own injury severity tends to vary widely and is subject to recall bias [7,8]. To date, there is no satisfactory surrogate for injury severity in LMICs. We need a validated tool to accurately measure injury severity in under-resourced communities not only to pursue burden-concordant resource allocation, but also to capture excess deaths to guide and monitor interventions and quality improvement efforts.

To address this gap, a targeted question set—the community-based injury severity (CBS) test—was developed and tested among lay community members in Cameroon. An affirmative response to any of the questions was considered to be a positive CBS test result, potentially indicative of more severe injury. The primary goal of this study was to prospectively cross-validate the CBS test with objective injury severity metrics available in the formal care setting in Cameroon using multi-site, hospital-based trauma registry data.

## Methods

This study encompasses the development, evaluation and validation of a four-question injury severity screening tool (Fig 1). The feasibility of the CBS test in both community and hospital-based settings was intrinsically assessed as well. We will first briefly describe the CBS question development. The succeeding methodology is bipartite: Part 1 describes the initial community-based evaluation of the CBS questions in a cross-sectional household-based survey on injury in the Southwest region of Cameroon [6]. Part 2 describes the subsequent prospective, multi-center cross-validation of the CBS questions in injured patients presenting for formal hospital care using objective metrics ascertained from four hospital-based trauma registries in Cameroon. Ethical approval was obtained from institutional review boards at the University of California, San Francisco (Part 1: IRB#15–18424, Part 2: IRB#13–12535), the University of California, Los Angeles (Part 2: IRB#19–000086), the University of Douala (Part 1: N˚-IEC-UD/694/10/2016/A) and the Cameroon National Ethics Committee (Part 2: N˚2014/09/496/CE/CNERSH/SP, N˚2018/09/1094/CE/CNERSH/SP). Ethical review for the CBS question development was encompassed under the review for "Part 1." All participants were verbally consented by Cameroonian research assistants using a standardized oral script in the participants' native language. For Part 1, participants signed their name to indicate consent. For Part 2, consent was entirely verbal. Like with United States-based trauma registries, consent was obtained for Part 2 after the patient was stabilized. If the patient did not give consent, their

## Developing the CBS Questions

1. Iterative question development by local physicians
2. Face validity evaluated with local families
3. Questions pilot-tested in the community

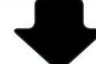

## Part 1. Evaluating the CBS Test in the Community

Cross-Sectional Mixed-Methods Community-Based Survey (Jan - Mar 2017)

| *Per Household:* Demographic & Socioeconomic Information | *Per Injury:* • Basic injury characteristics • Care-seeking behavior, hospitalization • Outcomes: Disability, death • Economic impact | *Per Injury:* CBS Questions |
|---|---|---|

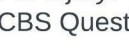

## Part 2. Cross-Validating the CBS Test Using Hospital-Based Trauma Registry Data

Prospective Multi-Center Validation Study (Oct 2017 - Dec 2019)

| *Per Patient:* Demographic & Socioeconomic Information | *Per Patient:* • Physician-identified injury characteristics • Anatomic & physiologic variables • Diagnostic & treatment variables • Outcomes: Hospitalization, LOS, death | *Per Patient:* CBS Questions |
|---|---|---|

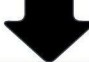

**Fig 1. Study methodology.**

data was withdrawn and destroyed. The consent process as described for both Part 1 and Part 2 was approved by both local and United States-based ethical review boards. Additional information regarding the ethical, cultural, and scientific considerations specific to inclusivity in global research is included in the (S1 Checklist).

## CBS question development

The CBS indicator questions were developed iteratively with Cameroonian physicians and lay-persons. For initial question development, we queried local surgeons on symptoms with corresponding questions likely indicative of severe injury. Questions were then iteratively revised with expert insight until consensus. We then pre-tested the proposed questions on lay Cameroonian nationals for community use in two main steps. First, research assistants tested potential questions on family members to ensure comprehensibility, acceptability and face validity

In the 24 hours post-injury…
1. Did the injured individual experience loss of consciousness?
2. Did they stop breathing?
3. Were they confused or disoriented after the event?
4. Did they have amnesia about the event?

**Fig 2. Community-Based Injury Severity (CBS) questions.**

with resultant modification. Next, we pilot-tested the questions on 200 cluster-selected community members and then further refined them (Fig 2). Of note, these community members were drawn from clusters that were not selected in Part 1 (below).

## Study design, population & sampling

**Part 1.** The initial community-based evaluation of the CBS questions was part of a larger pilot-tested, mixed-methods cross-sectional household survey on injury. The survey was conducted in 2017 in southwest Cameroon, a mixed rural-urban region of 1.5 million [9]. The three-stage cluster-sampling methodology, powered to adequately capture traumatic injury and death in the community, is described elsewhere [6].

In addition to demographic and socioeconomic household data, this survey collected information on each household member, living or deceased, who, in the past 12 months, had sustained an injury resulting in death, loss of routine activity, or need for medical attention, irrespective of whether formal care was obtained. For each injury identified, further information on injury characteristics, care-seeking behavior, treatment, disability, and economic ramifications was collected. Respondents were also asked to answer the four CBS questions for each injury event.

**Part 2.** After evaluating the performance of the CBS questions in the community, we prospectively cross-validated the questions at four Cameroonian hospitals in the Southwest and Littoral regions: Limbe Regional Hospital (Southwest), Laquintinie Hospital of Douala (Littoral), Catholic Hospital of Pouma (Littoral), and Edea Regional Hospital (Littoral). While Pouma is a faith-based, non-profit hospital, the other three are public, regional referral (i.e., secondary or tertiary) hospitals. Each began participating in a coordinated trauma registry (the Cameroon Trauma Registry) in June 2015, with the exception of Edea (January 2016). Until January 2020, all patients presenting with an injury, regardless of admission, were included in the respective hospital's trauma registry (post-2020, only patients who were deceased, admitted, transferred or discharged against medical advice were included in the registry).

Sample size calculations were targeted to provide ten affirmative and negative responses for each CBS variable, as well as the outcome of hospital death. Notably, given the potential of the first four questions to skew towards identifying traumatic brain injuries, a fifth potential CBS question ("Was the individual carried off scene?") was collected in the hospitals' trauma registries for further evaluation.

## Data collection & analysis

**Part 1.** Cameroonian research assistants trained in survey techniques collected household data from an adult household representative. Quantitative data was recorded on paper forms and then manually entered into an encrypted REDCap database. Statistical analysis was performed using STATA, version 14 (StataCorp LLC) and included 1) descriptive statistics

reporting on frequency analysis, 2) association analysis, reporting on Pearson chi-squared or Kruskal-Wallis, given non-parametric data, and 3) multiple logistic and linear regression analysis. We explored the association between the presence of CBS indicators and injury patterns, such as mechanism and anatomic location of injury, and outcomes known to be associated with injury severity, including days of disability, economic hardship, hospitalization and death. The relationship between CBS indicators and injury outcomes, such as days of disability was evaluated using multiple linear regression, adjusting for variables known to independently affect injury severity (e.g., age, formal care use, mechanism).

**Part 2.** Each hospital integrated the CBS questions into their respective trauma registry in October 2017. Upon hospital presentation, research assistants ask the patient or patient's companion the CBS questions, in addition to collecting routine trauma registry data. Routine trauma registry data includes patient demographics, injury characteristics, clinical variables, diagnostic and treatment interventions, and hospital outcomes, including mortality. Research assistants collaborate with the hospital registrar and emergency department staff to record all trauma registry data for each patient on paper forms. Registry data is subsequently entered electronically into an encrypted REDCap database. The Cameroon Trauma Registry has a field supervisor responsible for data integrity and quality assurance who cross-references ten percent of electronic records with paper forms at each hospital.

We extracted trauma registry data for all patients who presented for injury to the four hospitals between October 3, 2017 and December 31, 2019. Data was analyzed using STATA, version 16.1 (StataCorp LLC). The statistical analysis included 1) descriptive statistics reporting on frequency analysis, 2) association analysis, reporting on Pearson chi-squared or Kruskal-Wallis statistics, given non-parametric data, 3) multiple logistic regression, reporting on odds ratios (OR), and 4) receiver-operating characteristic (ROC) analysis, reporting on area under the curve (AUC). All reported confidence intervals (CI) are 95% confidence intervals and results with p-values less than 0.05 were considered to be statistically significant.

We explored the association between the presence of CBS indicators and potential injury outcome mediators, including socioeconomic factors, injury type, and pre-hospital patterns. We also examined the association between CBS indicators and physiologic variables indicative of severe injury. We used multiple logistic regression to evaluate the effect of age, sex, all socioeconomic variables, injury type, and presence or absence of the CBS indicators on mortality and hospital admission. The performance of the CBS indicators was compared to validated trauma severity scores, including the Revised Trauma Score (RTS) [10], the Kampala Trauma Score (KTS) [11], the GCS, Age and Pressure (GAP) score [12], and the Highest Estimated Abbreviated Injury Severity (HEAIS) score, using ROC analyses. The predictive ability of the CBS test for the outcome of death was analyzed similarly. CBS test performance was analyzed both 1) with and without the fifth potential indicator and 2) as a dichotomous (CBS indicators present or absent) versus continuous variable (number of CBS indicators present) to understand if the CBS test possessed cumulative predictive ability. Ultimately, the below analyses were performed using the dichotomous version of the CBS test with the original four indicators unless explicitly stated otherwise (rationale below).

## Results

**Part 1.** Of the 1551 households approached, 1287 households (83%) were ultimately surveyed, capturing information on 8065 individuals. Data was collected from 15 rural and 18 urban sampling areas. The study population had a median age of 24 and was 52% women. Demographics were representative of data from the 2011 Demographic and Health Survey [9]. 471 of the 8065 participants (5.8%) had sustained at least one injury in the past year, but some

**Table 1. Frequency of CBS indicators in community-based injuries.**

| Discrete Injury Events | Indicator Frequency n = 503 |
|---|---|
| | % |
| ≥1 Severity Indicator | 16.5% |
| Lost Consciousness | 8.0% |
| Stopped Breathing | 0.4% |
| Disorientation | 9.4% |
| Amnesia | 1.8% |

had been injured more than once, resulting in 503 discrete injuries in the past year (63 injuries/1000 person-years) with nine fatalities.

Among 503 injuries, 16.5% resulted in the presence of at least one severity indicator (Table 1). The presence of at least one severity indicator correlated with injury patterns and outcomes in the community. Specifically, severity indicators were more common after road traffic injury (OR 4.59, p<0.0001) and in those with head and neck (OR 7.15, p<0.0001) or torso (OR 3.83, p = 0.029) injury. Among those who presented to formal care, individuals with CBS indicators had higher hospitalization rates (50.0% vs. 26.5%, p = 0.004) and longer admissions (11.6 vs. 2.9 days, p = 0.03). Survivors with severity indicators were more likely to report ongoing disability (OR 1.9, p = 0.004) and, even after adjusting for age, formal care use, mechanism and anatomic region of injury, CBS indicators still predicted increased disability days (OR 23, p = 0.02).

**Part 2.** 9596 injured patients presented to the four trauma hospitals between October 3, 2017 and December 31, 2019. The presence or absence of CBS indicators was unknown in 21 patients (0.23%) and they were excluded from the analysis; 9575 patients were ultimately analyzed.

The median age of trauma registry patients was 30 and nearly 70% were male (Table 2). On average, individuals with CBS indicators (CBS+) were slightly older with greater male predominance (77.8%). They were also of lower socioeconomic status: Fewer were college-educated, urban or used liquid petroleum gas as their primary fuel source, a proxy for increased socioeconomic status in rural and peri-urban Cameroon [13]. Home or land ownership was associated with rural status.

**Table 2. Demographic & socioeconomic characteristics of registry patients.**

| | All Patients | # Responses | CBS+ Patients | CBS- Patients | |
|---|---|---|---|---|---|
| Total | n = 9575 | n = 9575 | n = 963 | n = 8612 | |
| | % | n | % | % | p |
| Age | 30 (22–40)[a] | 9516 | 31 (23–41)[a] | 30 (22–40)[a] | 0.0004* |
| Male Sex | 69.8% | 9522 | 77.8% | 69.4% | <0.0001* |
| College-Educated | 10.9% | 8851 | 9.2% | 11.1% | 0.081 |
| Urban | 88.7% | 9544 | 79.1% | 90.1% | <0.0001* |
| Own Their Home | 21.5% | 9331 | 24.1% | 21.9% | 0.113 |
| Own Cell Phone | 93.1% | 9482 | 92.2% | 93.5% | 0.139 |
| Own Agriculture Land | 14.3% | 9208 | 20.7% | 13.8% | <0.0001* |
| Primary Fuel: LPG[b] | 57.6% | 9086 | 51.2% | 58.3% | <0.0001* |

[a] Median (IQR).

[b] Liquid Petroleum Gas.

**Table 3. Injury type & pre-hospital characteristics of registry patients.**

| | All Patients | # Responses | CBS+ Patients | CBS- Patients | |
|---|---|---|---|---|---|
| **Total** | **n = 9575** | **n = 9575** | **n = 963** | **n = 8612** | |
| | % | n | % | % | p |
| Injury Type: | | 9477 | | | <0.0001* |
| Penetrating | 11.8% | | 6.1% | 12.4% | |
| Blunt | 82.8% | | 84.8% | 82.6% | |
| Burn | 1.6% | | 2.4% | 1.6% | |
| Other | 3.8% | | 6.7% | 3.5% | |
| Transport Distance (in km) | 6 (3–10)[a] | 8820 | 8(5–12)[a] | 5.5(3–10)[a] | 0.0001* |
| Transport Method: | | 9370 | | | <0.0001* |
| Police car or Ambulance | 4.3% | | 13.3% | 3.3% | |
| Motorcycle, Taxi, or Car | 87.7% | | 79.7% | 88.6% | |
| Walked In | 1.8% | | 0.2% | 1.9% | |
| Received Care at Scene | 14.2% | 9407 | 20.7% | 13.6% | <0.0001* |

[a] Median (IQR).

While more than 80% of injuries were blunt trauma, there were relatively fewer penetrating injuries in CBS+ individuals (6.1% vs. 12.4%, p<0.0001) (Table 3). A greater percentage of CBS+ individuals were transported by police car or ambulance and received care at the scene of the accident. They also traveled greater distances to the hospital.

*CBS Test Association with Clinical Variables.* Normal primary surveys were less common in CBS+ patients, reflected by patent airway (92.4% vs. 99.8%, p<0.0001), normal respirations (83.7% vs. 99.6%, p<0.0001) and palpable pulse (94.8% vs. 99.8%, p<0.0001) (Table 4). CBS + patients were also more likely to be tachycardic (34.3% vs. 28.0%, p<0.0001), hypotensive (7.1% vs. 1.9%, p<0.0001), and have a GCS less than nine (18.1% vs. 0.4%, p<0.0001). For patients with a second (n = 407) and third (n = 113) recorded set of vitals, CBS positivity was associated with a significantly lower median systolic blood pressure (SBP) on both subsequent readings (2nd SBP: 113 vs. 121, p = 0.0013; 3rd SBP: 103.5 vs. 120, p = 0.0131). Although only 53 patients received CPR and only 3 underwent invasive airway management (endotracheal intubation or cricothyrotomy), the preponderance of these patients were CBS+ (2.5% vs. 0.4%, p<0.0001 and 0.3% vs. 0.0%, p<0.0001, respectively). A greater proportion of CBS+ patients had evidence of external bleeding (77.3% vs. 65.2%, p<0.0001) and received a blood transfusion (6.3% vs. 1.6%, p<0.0001). The majority of registry patients were discharged home, but only one-quarter of CBS+ patients were discharged home concordant with medical advice. Far more fatalities occurred in CBS+ patients (18.5% vs. 0.6%, p<0.0001).

*Utility of CBS Test to Predict Mortality & Hospital Admission.* Age, use of LPG fuel, burn injuries, and the presence of at least one CBS indicator all significantly predicted mortality (p<0.05). Age minimally predicted death (OR 1.02, CI: 1.01–1.03, p<0.0001), while the use of LPG fuel was protective (OR 0.43, CI: 0.31–0.61, p<0.0001). Neither penetrating nor blunt injury correlated with increased death, but burns predicted increased mortality (OR 5.08, CI: 2.07–12.47, p<0.0001). The strongest predictor of death was the presence of a CBS indicator: Patients with at least one CBS indicator had significantly higher odds of dying compared to those without any CBS indicators (OR 30.26, CI: 21.30–42.98, p<0.0001). The presence of CBS indicators similarly predicted hospital admission (OR 2.98, CI: 2.53–3.50 (p<0.0001), controlling for all socioeconomic metrics, age, sex and injury type. None of the injury severity scoring systems (KTS, RTS, GAP, HEAIS, CBS) correlated significantly with hospital length of stay.

**Table 4. Clinical characteristics & outcomes of registry patients.**

|  | All Patients | # Responses | CBS+ Patients | CBS- Patients |  |
| --- | --- | --- | --- | --- | --- |
| **Total** | ***n = 9575*** | ***n = 9575*** | ***n = 963*** | ***n = 8612*** |  |
|  | **%** | ***n*** | **%** | **%** | ***p*** |
| Primary Survey: |  |  |  |  |  |
| Patent Airway | 98.4% | 9516 | 92.4% | 99.8% | <0.0001* |
| Normal Respirations | 98.0% | 9515 | 83.7% | 99.6% | <0.0001* |
| Palpable Pulse | 98.5% | 9498 | 94.8% | 99.8% | <0.0001* |
| Vitals & Exam: |  |  |  |  |  |
| Heart Rate | 85 (75–96)[a] | 8204 | 90 (78–101)[a] | 84 (75–96)[a] | 0.0001* |
| Systolic Blood Pressure | 126 (115–138)[a] | 7975 | 122 (109–137)[a] | 127 (115–139)[a] | 0.0001* |
| Glascow Coma Scale | 15 (15–15)[a] | 9476 | 14 (10–15)[a] | 15 (15–15)[a] | 0.0001* |
| External Bleeding | 65.5% | 9452 | 77.3% | 65.2% | <0.0001* |
| Treatment Variables: |  |  |  |  |  |
| Received CPR[b] | 0.6% | 9520 | 2.5% | 0.3% | <0.0001* |
| Crichothyrotomy/ETT[c] | 0.03% | 9575 | 0.31% | 0.00% | <0.0001* |
| Received Fluid | 55.9% | 5451 | 98.3% | 98.1% | 0.638 |
| Received Blood Transfusion | 0.7% | 2898 | 6.3% | 1.6% | <0.0001* |
| Outcomes: |  |  |  |  |  |
| Discharged (No AMA[d]) | 63.1% | 9363 | 25.2% | 67.3% | <0.0001* |
| Hospital Admission | 14.5% | 9363 | 31.2% | 12.7% | <0.0001* |
| LOS[e] (for admitted patients) | 0 (0–1)[a] | 9274 | 0 (0–1)[a] | 0 (0–1)[a] | 0.8151 |
| Death | 2.4% | 9466 | 18.6% | 0.6% | <0.0001* |

[a] Median (IQR).

[b] Cardiopulmonary Resuscitation.

[c] Endotracheal Intubation.

[d] Against Medical Advice.

[e] Length of Stay.

This trend was consistent even after exclusion of same-day discharges and/or patients who left against medical advice or were transferred.

*CBS Test vs. Validated Injury Severity Scoring Systems.* ROCs were compared between validated injury severity scoring systems (KTS, RTS, GAP, HEAIS) relying on clinical and physiologic measures and the CBS test. The CBS test was comparable to the aforementioned injury severity scoring systems with an AUC of 0.8029 (Fig 3).

*CBS Test Characteristics.* The majority of trauma registry patients were carried off of the scene (55.4%) and this fifth potential indicator was associated with decreased mortality odds in hospital-based patients (OR: 0.4, p<0.0001). Although this fifth variable increased the sensitivity of the CBS test to 97.8% (from 77.0%) in the hospital population, it also significantly decreased the specificity to 38.6% (from 91.6%). Of note, the CBS test improved slightly with only three questions compared to four, where exclusion of amnesia resulted in a slight increase in AUC with a small increase in specificity from 91.6% to 92.4% without impacting sensitivity. However, any further variable exclusion negatively affected the CBS test's predictive utility. This information may be useful to further refine the test and minimize data collection burden. ROC analyses show that the CBS indicators do not function additively to predict mortality; regardless of the number of CBS indicators, the dichotomous CBS test (presence or absence of CBS indicators) is a superior predictor of fatality. Specifically, when the original four CBS indicators and the potential five CBS indicators were each examined as independent continuous

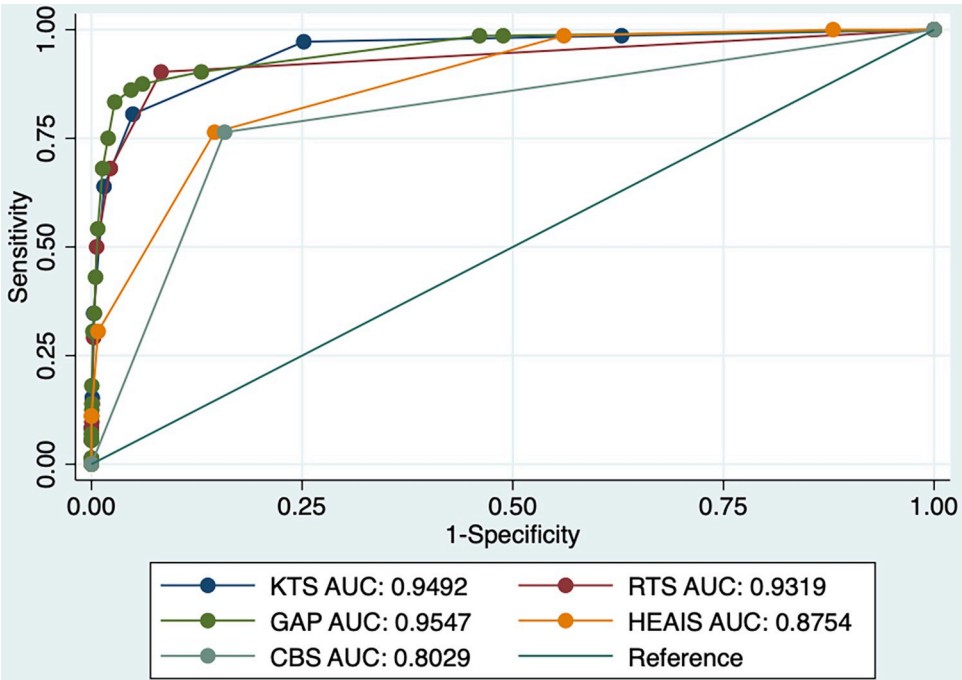

**Fig 3. ROC analysis of CBS test vs. other severity scoring systems (Event: Death).**

variables (e.g., CBS indicator score 0–4 and CBS indicator score 0–5, respectively), the AUC decreased.

*Data Availability for CBS Test vs. Validated Injury Severity Scoring Systems.* In this hospital-based study, only 24.7% of the 9575 registry patients had complete data to calculate all four validated scores (RTS, KTS, GAP, HEAIS). HEAIS was readily calculated in 97.5% of patients, but RTS and KTS were frequently unable to be calculated due to missing data. In particular, 63.7% of registrants were missing respiratory rate, impeding calculation of RTS or KTS. Data missingness was not random. Individuals missing critical score calculation data were more likely to be younger, female and urban. They were also less likely to receive care at the scene, blood transfusions, or CPR, or be admitted to the hospital. Death was also slightly less common in those with incomplete RTS, KTS, GAP or HEAIS scores (2.2% vs. 3.1%, p = 0.021).

## Discussion

In this study, we developed the CBS test, evaluated its performance in the community, and validated its ability to predict outcomes associated with increased injury severity in four hospital-based trauma registries in Cameroon. CBS test responses correlated with injury patterns and outcomes in the community indicative of severe injury and the test was found to be valid in hospital-based populations: The presence of CBS indicators (CBS+) was associated with increased clinical abnormalities and was highly-predictive of hospital admission and death. The CBS test offers us an essential marker of injury severity for community-based trauma epidemiology. Accurate trauma epidemiology can facilitate advocacy, informed organizational change, quality improvement monitoring, and pre-hospital care system development. Furthermore, the CBS test has potential as a triage tool in both the pre-hospital and hospital intake settings. One of the most salient characteristics of the CBS test is the ease of calculation in all contexts.

Sub-Saharan Africa, and Cameroon in particular, are leading the way in adapting traditional resource-intensive trauma severity scores to under-resourced contexts. Underreporting in low-resource settings stymies accurate traumatic injury estimates with myriad sequelae [14]. Even in hospitals and clinics, adequate physiologic information is not consistently available to calculate many established trauma severity scores, likely due to a shortage of equipment and trained personnel [15]. In this study, only one-quarter of registry patients had complete data to calculate all five scores, despite data derivation from a trauma registry of hospital-based patients supported by dedicated staff. Whereas HEAIS was readily calculated, a majority of patients were missing respiratory rate, an oft unrecorded variable in trauma registries, impeding calculation of RTS or KTS [16]. Difficulty measuring respiratory rate may be more specific to the trauma setting than the resource-constrained setting, where both patient factors (e.g., agitation, altered mental status) and treatment factors (concurrent rapid assessment and treatment) complicate reliable respiratory rate measurement [17]. Previously validated injury severity scores may demonstrate slightly superior death discrimination but these scores are often impractical in under-resourced injury settings. For epidemiologic studies, the CBS test can also be applied retrospectively. The CBS test predicts injury severity remarkably well using simple, readily-available information.

The CBS test fills an important gap in trauma epidemiology in under-resourced settings. Timely deployment is warranted: Africa is projected to have the fastest urban growth rate and an additional 950 million people are projected to inhabit its cities by 2050 [18,19]. The African continent already has the highest road traffic injury fatality rate [20] with models indicating the fatality rate may be even higher than reported [14]. With urbanization, road traffic and its associated injuries will only further intensify [19]. Traumatic injury is often overlooked by the international community and is severely underfunded, receiving only $0.04 per disability-adjusted life year (DALY) in global development assistance compared to tuberculosis ($25.09/DALY) and HIV ($4.05/DALY) [18]. Furthermore, 62% of trauma care development assistance is spent on war victims accounting for 3% of the injury burden [18]. Sub-Saharan Africa, in particular, receives minimal trauma care assistance [18]. Accurate traumatic injury estimates will facilitate domestic and international advocacy for burden-concordant attention and resources.

Given impending African urbanization, it is imperative to begin planning efficient systems of traumatic injury prevention and care [19]. Country assessments identify enormous gaps in injury care but evidence from higher-resource settings indicates that mortality can be significantly reduced with organizational and administrative change, from pre-hospital triage to accreditation of trauma care facilities [21]. In Cameroon, local injury research and planning has already begun. Mapping and characterizing injury in communities can directly inform targeted prevention and care strategies, particularly pre- and post-hospital care. Although we cannot establish etiology from cross-sectional data, CBS+ populations most likely represent groups in which there is a higher risk of severe injury. Injury patterns have been used extensively for focused harm-reduction strategies, especially road traffic injuries [22].

Characterizing traumatic injury in the community has the potential to bolster deliberate development of incipient pre-hospital care systems. The CBS questions show promise as a triage tool in both the community and hospital intake settings, particularly when trained personnel and basic diagnostic equipment are lacking. In areas without formal pre-hospital systems, lay first responder training in trauma care is recommended, costing only $0.12 per capita [23]. There is potential to coordinate lay taxi drivers to develop a rudimentary system of pre-hospital transport. In order to understand the quality and impact of interventions at all levels, we must accurately capture injury severity to quantify changes in excess morbidity and mortality.

Notably, although we compared the diagnostic accuracy (AUC) of the CBS test to that of hospital-based injury severity scores, the conditions of application are distinct. The CBS test is

intended to be applied retrospectively in the community for accurate trauma epidemiology or prospectively in settings where trained personnel or basic diagnostic equipment are not readily available. As with any test, the adequacy of the CBS test's sensitivity (77.0%) and specificity (91.6%) depend on the consequences of misclassification (i.e., false-positives and false-negatives). Although the accuracy of untrained, lay personnel in identifying severe injury is unknown, the CBS test identifies severe injury far more accurately than no test (AUC: 0.5). Conventional injury severity scoring systems may have slightly higher mortality discrimination but these scores cannot be applied in the community setting. The CBS test is not intended to replace the need for formal trauma evaluation by a trained clinician in an appropriately-equipped, hospital-based setting.

As noted above, it may be appropriate to further simplify the CBS test to three questions to minimize data collection burden, but this warrants further validation; in patients presenting acutely post-trauma, there may have been inadequate time between the injury event and hospital presentation for amnesia to become noticeable. The CBS test may also benefit from further evaluation of its ability to predict future disability in formal care recipients. Accordingly, mHealth technologies are currently being deployed in Cameroon to improve post-traumatic injury outpatient follow-up. Additional information on ongoing disability in previously-hospitalized, registered trauma patients can be used to further understand the CBS test's ability to predict non-fatal injury severity.

## Limitations

One key potential limitation is the CBS test's propensity to identify severe traumatic brain injury. Like scoring systems that incorporate GCS, it can be difficult to disaggregate altered mental status as a sequela of shock and poor neurologic perfusion from direct injury to the brain. However, the low rates of CPR, blood transfusion and death in CBS-negative patients suggest that the CBS test possesses adequate sensitivity in patients without traumatic brain injury. Another potential limitation of this study is the lack of adjustment for secular trends given the study timing. Because hospital-based validation of the CBS test began after completion of the community-based evaluation, there is no way to assess the impact of improvements in traumatic injury care on CBS test performance. Although the CBS test can be applied retrospectively, results may be subject to recall bias. Finally, because the CBS test was evaluated in hospital-based trauma patients who have a higher prevalence of severe injury, the positive predictive value (PPV) and negative predictive value (NPV) of the CBS test in this setting should not be extrapolated to the general population. We have not reported on PPV or NPV in this study for this reason.

## Conclusion

A simple surrogate metric for injury severity is needed in under-resourced settings for accurate injury epidemiology, including excess mortality. The CBS test can be readily calculated in all contexts, in contrast to other injury severity scores, and is highly predictive of mortality. CBS-positivity is also associated with significant physiologic abnormalities indicative of the injury severity spectrum between wellness and death. The CBS test can be deployed widely in community-based surveys to improve estimates of injury severity and shows promise as an independent severity screening tool in the pre-hospital and hospital settings as well.

## Supporting information

**S1 Checklist. Inclusivity in global research.**
(DOCX)

**S1 Table. Frequency of CBS indicators in hospital-based injuries.**
(DOCX)

**S1 Data.**
(XLSX)

## Acknowledgments

We would like to acknowledge the Ministry of Health in Cameroon, the research assistants for the community-based injury survey, and the trauma registry team at each participating hospital. Without their dedication, many aspects of this work, particularly data collection, would not be possible.

## Author Contributions

**Conceptualization:** Rasheedat Oke, Catherine Juillard, Alain Chichom-Mefire, S. Ariane Christie.

**Data curation:** Rasheedat Oke, Catherine Juillard, Fanny Dissak-DeLon, Alain Chichom-Mefire, Mbiarikai Agbor Mbianyor, Georges Alain Etoundi-Mballa, Thompson Kinge, Louis Richard Njock, Daniel N. Nkusu, Jean-Gustave Tsiagadigui, S. Ariane Christie.

**Formal analysis:** Haley Tupper, Mark Yost, S. Ariane Christie.

**Funding acquisition:** Rasheedat Oke, Catherine Juillard, S. Ariane Christie.

**Investigation:** S. Ariane Christie.

**Methodology:** Rasheedat Oke, Catherine Juillard, Alain Chichom-Mefire, S. Ariane Christie.

**Project administration:** Haley Tupper, Rasheedat Oke, Catherine Juillard, Fanny Dissak-DeLon, Alain Chichom-Mefire, Mbiarikai Agbor Mbianyor, Georges Alain Etoundi-Mballa, Thompson Kinge, Louis Richard Njock, Daniel N. Nkusu, Jean-Gustave Tsiagadigui, Melissa Carvalho, S. Ariane Christie.

**Resources:** Rasheedat Oke, Fanny Dissak-DeLon, Alain Chichom-Mefire, Mbiarikai Agbor Mbianyor, Georges Alain Etoundi-Mballa, Thompson Kinge, Louis Richard Njock, Daniel N. Nkusu, Jean-Gustave Tsiagadigui, Melissa Carvalho.

**Supervision:** Catherine Juillard, S. Ariane Christie.

**Validation:** Haley Tupper, S. Ariane Christie.

**Visualization:** Haley Tupper.

**Writing – original draft:** Haley Tupper.

**Writing – review & editing:** Haley Tupper, Rasheedat Oke, Fanny Dissak-DeLon, Mark Yost, S. Ariane Christie.

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
