## [Decision Letter · Decision Letter 0]

19 Apr 2023

PGPH-D-23-00504

The CBS test: Development, evaluation & cross-validation of a community-based injury severity scoring system in Cameroon

Dear authors,

Thank you for submitting your manuscript to PLOS Global Public Health. After careful consideration, we feel that it has merit but does not fully meet PLOS Global Public Health’s publication criteria as it currently stands. Therefore, we invite you to submit a revised version of the manuscript that addresses the points raised during the review process.

We look forward to receiving your revised manuscript.

Kind regards,

Andreas K Demetriades, MBBChir, MPhil, FRCSEd, FEBNS.

Academic Editor

Journal Requirements:

2. Please send a completed 'Competing Interests' statement, including any COIs declared by your co-authors. If you have no competing interests to declare, please state "The authors have declared that no competing interests exist". Otherwise please declare all competing interests beginning with the statement "I have read the journal's policy and the authors of this manuscript have the following competing interests:"

3. Please amend your detailed Financial Disclosure statement. This is published with the article. It must therefore be completed in full sentences and contain the exact wording you wish to be published.

a. State what role the funders took in the study. If the funders had no role in your study, please state: “The funders had no role in study design, data collection and analysis, decision to publish, or preparation of the manuscript.”

b. If any authors received a salary from any of your funders, please state which authors and which funders.

Additional Editor Comments (if provided):

Reviewers' comments:

Reviewer's Responses to Questions

**Comments to the Author**

1. Does this manuscript meet PLOS Global Public Health’s publication criteria? Is the manuscript technically sound, and do the data support the conclusions? The manuscript must describe methodologically and ethically rigorous research with conclusions that are appropriately drawn based on the data presented.

Reviewer #1: Yes

Reviewer #2: Yes

2. Has the statistical analysis been performed appropriately and rigorously?

Reviewer #1: Yes

Reviewer #2: Yes

3. Have the authors made all data underlying the findings in their manuscript fully available (please refer to the Data Availability Statement at the start of the manuscript PDF file)?

Reviewer #1: Yes

Reviewer #2: Yes

4. Is the manuscript presented in an intelligible fashion and written in standard English?

Reviewer #1: Yes

Reviewer #2: Yes

5. Review Comments to the Author

Reviewer #1: Overall, this was a very interesting study that provides a simple trauma score framework that may be deployable in resource-limited settings. It will be interesting to see how the CBS score can be further refined in future studies.

Out of curiosity, why did you look at fuel sources? This was unclear in the manuscript.

Reviewer #2: Authors present the development and validation of the CBS, a novel injury severity scoring system, to be used in injury surveillance in low- and middle-income settings. They present clear, thorough analysis and findings, highlighting that CBS performs sufficiently similarly to other, more complex scoring systems and building a narrative that supports further research towards adoption of CBS in this setting. The paper makes a significant contribution to the literature and may be further strengthened by answering the following questions in the text.

Major comments:

1. What does looking at the “presence or absence of CBS indicators” as a variable indicate with respect to the objective of this study? Please outline this connection explicitly. It wasn’t immediately clear that this is the variable of interest.

2. Is the decrease in AUC acceptable to justify the use of CBS in this setting over other options? Under what conditions or justification? Please outline this to support your claim that the CBS test is a viable option for triage.

3. Could you speculate on why the CBS+ and CBS- populations are different? What contributes to this? What would potential adopters of this scoring system need to think about?

Minor comments:

1. What happened to trauma registry data collection after January 2020? Did something make data collection no longer possible? This would be helpful to outline if it gives context on feasibility of such data collection into the future. What needs to be considered here for something to work?

2. You do more than simply cross-validate CBS here. You also look at (and make claims for) its feasibility of implementation. I would consider adding this point when you outline the study objective to set readers’ expectations.

3. Given the number of potential injury severity scores, why do you think one of them hasn’t been picked up and implemented regularly in the Cameroonian context? Similar efforts, like for KTS, have been carried out, and you suggest that even KTS is not appropriate here. In the introduction, it may be helpful to highlight issues specific to Cameroon that make existing approaches infeasible. For example, why does a community-based score work better than KTS, which was created for resource-constrained settings?

6. PLOS authors have the option to publish the peer review history of their article (what does this mean?). If published, this will include your full peer review and any attached files.

**Do you want your identity to be public for this peer review?** For information about this choice, including consent withdrawal, please see our Privacy Policy.

Reviewer #1: **Yes: **Vinay N Kampalath

Reviewer #2: No

---

## [Decision Letter · Decision Letter 1]

8 Jun 2023

The CBS test: Development, evaluation & cross-validation of a community-based injury severity scoring system in Cameroon

PGPH-D-23-00504R1

Dear authors,

We are pleased to inform you that your manuscript 'The CBS test: Development, evaluation & cross-validation of a community-based injury severity scoring system in Cameroon' has been provisionally accepted for publication in PLOS Global Public Health.

Best regards,

Andreas K Demetriades, MBBChir, MPhil, FRCSEd, FEBNS.

Academic Editor

Congratulations; the peer review process recommends acceptance of your submitted manuscript.

Reviewer Comments (if any, and for reference):

Reviewer's Responses to Questions

**Comments to the Author**

1. If the authors have adequately addressed your comments raised in a previous round of review and you feel that this manuscript is now acceptable for publication, you may indicate that here to bypass the “Comments to the Author” section, enter your conflict of interest statement in the “Confidential to Editor” section, and submit your "Accept" recommendation.

Reviewer #1: All comments have been addressed

Reviewer #2: All comments have been addressed

2. Does this manuscript meet PLOS Global Public Health’s publication criteria? Is the manuscript technically sound, and do the data support the conclusions? The manuscript must describe methodologically and ethically rigorous research with conclusions that are appropriately drawn based on the data presented.

Reviewer #1: Yes

Reviewer #2: Yes

3. Has the statistical analysis been performed appropriately and rigorously?

Reviewer #1: Yes

Reviewer #2: Yes

4. Have the authors made all data underlying the findings in their manuscript fully available (please refer to the Data Availability Statement at the start of the manuscript PDF file)?

Reviewer #1: Yes

Reviewer #2: Yes

5. Is the manuscript presented in an intelligible fashion and written in standard English?

Reviewer #1: Yes

Reviewer #2: Yes

6. Review Comments to the Author

Reviewer #1: Thank you for your reply to the comments. I believe you've adequately responded to feedback, and the additional clarification (particularly about the AUC) is very helpful. This work is an important contribution to trauma and emergency care literature.

Reviewer #2: Thank you for taking the time to so thoroughly address my questions and comments.

Congratulations on the work and this contribution to the field.

7. PLOS authors have the option to publish the peer review history of their article (what does this mean?). If published, this will include your full peer review and any attached files.

**Do you want your identity to be public for this peer review?** For information about this choice, including consent withdrawal, please see our Privacy Policy.

Reviewer #1: **Yes: **Vinay Kampalath

Reviewer #2: No
